# NPR1 and Redox Rhythm: Connections, between Circadian Clock and Plant Immunity

**DOI:** 10.3390/ijms20051211

**Published:** 2019-03-10

**Authors:** Jingjing Zhang, Ziyu Ren, Yuqing Zhou, Zheng Ma, Yanqin Ma, Dairu Hou, Ziqin Xu, Xuan Huang

**Affiliations:** 1Provincial Key Laboratory of Biotechnology of Shaanxi, Xi’an 710069, China; twinklezhang@stumail.nwu.edu.cn (J.Z.); reneeziyu@stumail.nwu.edu.cn (Z.R.); desertree0302@163.com (Y.Z.); 0mazheng0@stumail.nwu.edu.cn (Z.M.); mayanqin@stumail.nwu.edu.cn (Y.M.); dairuhou@stumail.nwu.edu.cn (D.H.); ziqinxu@nwu.edu.cn (Z.X.); 2College of Life Sciences, Northwest University, Xi’an 710069, China; 3Key Laboratory of Resource Biology and Biotechnology in Western China, Ministry of Education, Xi’an 710069, China

**Keywords:** plant immunity, circadian clock, SA-signaling network, NPR1, redox rhythm, NADH/nicotinamide adenine dinucleotide phosphate (NADPH), reactive oxygen species (ROS)

## Abstract

The circadian clock in plants synchronizes biological processes that display cyclic 24-h oscillation based on metabolic and physiological reactions. This clock is a precise timekeeping system, that helps anticipate diurnal changes; e.g., expression levels of clock-related genes move in synchrony with changes in pathogen infection and help prepare appropriate defense responses in advance. Salicylic acid (SA) is a plant hormone and immune signal involved in systemic acquired resistance (SAR)-mediated defense responses. SA signaling induces cellular redox changes, and degradation and rhythmic nuclear translocation of the non-expresser of *PR* genes 1 (NPR1) protein. Recent studies demonstrate the ability of the circadian clock to predict various potential attackers, and of redox signaling to determine appropriate defense against pathogen infection. Interaction of the circadian clock with redox rhythm promotes the balance between immunity and growth. We review here a variety of recent evidence for the intricate relationship between circadian clock and plant immune response, with a focus on the roles of redox rhythm and NPR1 in the circadian clock and plant immunity.

## 1. Introduction

Most eukaryotic and some prokaryotic organisms have some type of circadian clock that functions as an internal timing mechanism for the control of various normal physiological reactions under light/dark cycle, such as metabolism, hormone levels, body temperature, permitting vision, and brain activities processes [1,2,3,4,5]. The circadian clock is a rather robust system, especially in detecting the precise time of day processes in many metabolic and physiological reactions to help plants make the right reaction at the right time. Furthermore, it drives an endogenous oscillating rhythm strictly observe a period of approximately 24 h even if it is under continuous light or dark [6,7,8,9]. Diverse physiological pathways are connected with the circadian clock, including ion homeostasis, hormone signaling, sugar sensing, photoperiodic flowering and stress signaling. Increasing evidence has proved that plant immune response are also connected with the circadian clock [10,11,12]. Preserving circadian clock regulation during the day increases carbon dioxide fixation, growth, and fitness [13,14]. In addition, at least 30% of genes expression with robust oscillations in *Arabidopsis* are associated with physiological rhythms driven by the circadian clock [15].

Plants have evolved highly efficient defense mechanisms against a variety of major pathogens (*e.g*., *Pseudomonas syringae*, *Golovinomyces cichoracearum*, *Botrytis cinerea*, *Alternaria brassicicola*) [16,17,18]. There are two “branches” of triggered immunity in plants. (i) PAMP/MAMP-triggered immunity (PTI/MTI) is triggered by conserved pathogen or microbe-associated molecular patterns (PAMPs, MAMPs) [19]. Specific pattern recognition receptors (PRRs), such as EF-Tu receptor (EFR) and FLAGELLIN-SENSING 2 (FLS2) [20], recognize PAMPs or MAMPs and trigger PTI/MTI for resistance to biotrophic cellular pathogens, viruses, and insects [21]. (ii) Effector-triggered immunity (ETI), which is triggered by specific recognition between pathogen effectors and immune receptors in a host cell, generally encompasses a programmed cell death at the site of pathogen recognition known as the hypersensitive response (HR), and the receptors encoded by resistance genes (*R* genes) [22,23,24]. Subsequently (hours or days later), systemic acquired resistance (SAR) occurred following HR, which evidently result from cooperative activation of numerous genes termed pathogenesis-related (*PR*) genes [25,26,27,28]. Salicylic acid (SA), a plant hormone involved in development and basal resistance against multiple pathogens, is a key regulator of pathogen-induced SAR [29,30,31,32,33,34].

Numerous studies indicate a relationship among circadian clock, plant immunity, and reduction-oxidation (redox) rhythm [35,36,37]. Accumulation of reactive oxygen species (ROS) induced by PTI may prevent pathogen invasion [38]. ETI induces programmed cell death associated with changes in nicotinamide adenine dinucleotide phosphate (NADPH) oxidase activity that has been evidenced as the source of ROS [39]. Non-expresser of *PR* genes 1 (*NPR1*, also known as *NIM1* and *SAI1*) [40] regulates expression of clock genes and is also involved in plant defense response through its role in SA signaling [41,42]. In coordination with circadian clock changes, SA regulates expression of downstream defense genes in response to pathogen behavior [43]. In this article, we review interactions between circadian clock and plant immune system, we focus on recent advances in rhythmic oscillation of both redox and clock, and on metabolic and physiological reactions that promote balance of plant immunity and growth.

## 2. Relationship between Circadian Clock and Immune Responses

The core element of the circadian systems is an oscillator based on a transcription-translation negative feedback loop [44,45]. The circadian clock regulates plant innate immunity to invasion by exogenous pathogens. The coordination between biotrophic pathogens and host provides a daily schedule for plants to defend against exogenous pathogens, that plant defense may related to circadian clock. [6,8,46]. However, the link between them has never been firmly established.

### 2.1. The Core Loop of Circadian Clock

*A. thaliana* has a core loop structure of interrelated morning and evening loops which encodes molecular components of the circadian oscillator. Two partially redundant morning-phased genes, *CCA1* (*CIRCADIAN CLOCK ASSOCIATED 1*) [47] and *LHY* (*LATE ELONGATED HYPOCOTYL*) [48] have transcript and protein levels that peak in the morning and activate expression of *PRR* family genes (*PRR5*, *PRR7*, and *PRR9*/NI; *PSEUDO-RESPONSE REGULATORs 5*, *7*, *9*/night inhibitor) through direct association with their promoters [49,50]. *PRR* family genes also suppress *CCA1* and *LHY* expression by binding to their promoters. *PRR1*, the eponymous member of the *PRR* family, is also referred to as evening-phased *TOC1* (*TIMING OF CAB 1*) [51,52]. In antiphase to *TOC1* peak in early evening, ~40% of TOC1 target genes have an early morning phase regulated by circadian clock [53].

Loss of core loop elemental genes notably speeds up the clock [54]. Feedback loops have been shown to form the circadian clock and to regulate rhythms of output genes whose expression is reflected in behavioral and physiological rhythms in response to environmental changes throughout the day [55,56]. LHY and CCA1 have both been shown to regulate *TOC1*, *LHY*, and *CCA1* gene expression negatively (Figure 1) [47,56,57]. TOC1 acts as a repressor that directly binds to *CCA1* and *LHY* promoters in early evening [58,59]. Mutations of *CCA1* (*cca1-1*), *LHY* (*lhy-20*) and *TOC1* (*toc1-2*) did not completely disrupt clock rhythm. Loss-of-function *Arabidopsis* mutants, in comparison with wild-type, showed shorter appropriate duration of circadian rhythms and reduced fitness [13,52,60,61,62]. Semi-dominant *toc1-1* mutant revealed the strong effect of TOC1 on clock-controlled output processes (Table 1) [63]. CCA1 and LHY proteins evidently have analogous functions and, even if one of them is absent, the other is able to maintain circadian rhythm.

### 2.2. Regulation of Plant Immunity by Circadian Clock/Rhythm

Numerous studies have demonstrated direct regulation of plant immunity by circadian clock. PTI and ETI will be triggered through regulating evening-phased elements in plants involved in the circadian clock.

#### 2.2.1. *CCA1* Reduces Plant Susceptibility to Pathogens

Clock gene *CCA1* suppresses plant susceptibility to pathogens. It enhances pathogen resistance at dawn, as evidenced by changes in loss-of-function mutants of susceptibility to *Hyaloperonospora arabidopsidis* (*Hpa*) Emwa1, the cause of downy mildew disease in *Arabidopsis* leaves (Figure 1) [73,74]. *CCA1* encodes the transcription factor that contain a single MYB domain. Resistance to downy mildew was enhanced in a *CCA1* overexpression mutant (*CCA1*-ox) (Table 1) [47]. Resistance to pathogens at dusk is reduced in *CCA1* deletion mutants (Table 1) [75]; these plants show greater resistance in the morning and greater susceptibility in the evening. Rhythmic susceptibility changes throughout the day in *Arabidopsis* interaction with virulent biotrophic pathogen *P. syringae pv. tomato* DC3000 (DC3000), which triggers PTI, were not observed in *CCA1*-ox and *EARLY FLOWERING 3-1* mutants (*elf3-1*), indicating the ability of circadian clock to regulate the host immune response [76]. Defense response of ETI is more intense than that of PTI. Some plants recognize effector molecules secreted by pathogens through *R* genes, and then activate ETI to strengthen the defense response [30].

(1) *CCA1*, *LHY*, and *TOC1* form the core loop of transcription-translation feedback loops that regulate daily timekeeping. *CCA1* and *LHY* have transcript and protein levels that peak in the morning, *TOC1* peaks in early evening and acts as a repressor that directly binds to *CCA1* and *LHY* promoters. ~40% of TOC1 target genes have an early morning phase regulated by the circadian clock. LHY and CCA1 have both been shown to regulate *TOC1* negatively. (2) *CCA1* and *LHY* activate expression of *PRR* family genes through direct association with their promoters. *PRR* family genes also suppress *CCA1* and *LHY* expression by binding to their promoters. (3) *CCA1* and *LHY* regulate *GRP7*, a key constituent of a slave oscillator that modulates stomatal activity related to plant immune response, is involved in regulation of plant immunity. Stomatal opening at dawn and closure at noon. (4) Morning-phased *LHY* and *CCA1* positively regulate resistance against oomycete pathogen *Hpa* and bacterial pathogen *P. syringae*, whereas evening-phased *TOC1* negatively regulates resistance against bacteria. Spore formation of oomycete pathogen occurs mainly at night, and spores, therefore, germinate at dawn. (5) Maximal JA accumulation occurs around midday, whereas SA peaks around midnight, cytosolic NPR1 suppressed JA signaling. (6) NPR1 plays essential roles in binding of SA. NPR1 oligomers become monomers under pathogen infection, and this process triggers SA accumulation. In the absence of SA or pathogen challenge, NPR1 is degraded by the proteasome and its inhibitory effect on effector-triggered cell death and anti-pathogen defense is eliminated. (7) Under anti-pathogen defense response or SA induction, NPR1 is converted into monomers, which can be combined with TGA2 transcription factor to promote *PR* gene expression, and are released (transferred) from cytoplasmic sites to the nucleus. GSNO facilitates Cys156 disulfide bonding and promotes NPR1 oligomer formation. Both TRX-3h and TRX-5h catalyze monomerization of NPR1 and prevent its repolymerization.

Even in the absence of pathogen invasion, plants are programmed to deal with pathogen infection on the basis of a circadian clock schedule. Appropriate clock function plays a key role in defense responses. Spore formation of oomycete pathogen occurs mainly at night, and spores therefore germinate at dawn [77]. Defense response-related genes have adapted to such timing by peaking around midnight and dawn (Figure 1); an example is *CCA1*, overexpression of which enhances resistance in *Arabidopsis* defense response [47]. *Hpa* EmwaI defense in *Arabidopsis* stimulated not only programmed cell death but also basal resistance through *RECOGNITION OF PERONOSPORA PARASITICA 4* (*RPP4*), whose promoters show close association with the circadian clock. Expression of *RPP4* and its target genes shows broad overlap with CCA1 rhythm, CCA1 can promote the expression of the target defense genes related to programmed cell death (PCD) by activating RPP4 directly [64]. Certain other RPP4-target defense genes independent of *RPP4* can also be activated by regular changing CCA1 level for basal defense. The dual activation mode is resulting in a more powerful and efficient defense response [12].

#### 2.2.2. Does *CCA1* Regulate Non-Clock-Related Activities?

Does *CCA1* plays a role in regulation of non-clock-related activities for enhancement of anti-pathogen defense responses? As an output of the circadian clock, plant immune response is regulated by the clock under both constant light and circular lighting (12 hr light/12 hr dark). Under both these light conditions, susceptibility to avirulent *P. syringae* strain was greater for *CCA1*-ox plants than for wild-type, the opposite of this, *CCA1*-ox plants showed an enhanced resistance to *Hpa* [46,62]. Both *cca1-1/lhy-20* double mutant and *LHY*-ox plants showed enhanced susceptibility to avirulent *P. syringae*, and *cca1-1/lhy-20* double mutant also showed enhanced susceptibility to oomycete pathogen *Hpa* strains under circular lighting. Although *CCA1* and *LHY* mutants showed similarly shortened circadian period, *CCA1*-ox plants show enhanced immune response to *Hpa*, but LHY does not show a defense response against *Hpa* [64]. These findings suggest that the circadian clock regulates plant innate immunity as an output under both light conditions, however, whether the enhanced resistance to *Hpa* in *CCA1*-ox plants is caused by non-clock function resulting from *CCA1* overexpression is still unclear.

#### 2.2.3. Interactions between Plant Immunity and Circadian Clock/Rhythm

Mutual regulation between the circadian clock and plant immune response was observed in the clock-related gene mutants mentioned above, and further investigation showed that defense response was under the control of the circadian clock. When circadian clock rhythm resulting from *CCA1* or *LHY* overexpression was disrupted, susceptibility to *P. syringae* strain was greatly enhanced. *CCA1* and *LHY* regulate their downstream target gene *GLYCINE-RICH RNA-BINDING PROTEIN 7* (*GRP7*), which is involved in regulation of plant immunity and is a key constituent of a slave oscillator [78,79,80] that modulates stomatal activity related to plant immune response [81]. Interestingly, the circadian clock regulates plant defense response, and defense response in turn regulates the circadian clock. For example, treatment of avirulent *P. syringae* strain infection or of flg22 (a resulting flagellin protein) is associated with modulation of clock activity [62].

SA can be transiently produced upon infection [82]. A previous finding has shown that the amplitude of clock rhythms will increase under SA treatment [42]. In contrast to this, a lately study observed both SA treatment and *Pst* DC3000 infection decrease the amplitude of clock rhythms [83]. The sharp different result may caused by different SA concentrations or plants age [84,85]. Notably, SA treatment leads to a negative phase shift, which instead a slight phase advance in *Pst* DC3000 infection plants, only *Pst* DC3000 infection plants lengthened the clock period, which has no evident change under SA treatment [83]. Altogether, these data suggests that the ultimate clock phenotypes show in infection plants is due to SA-dependent and -independent pathways integration.

The stress- and defense-related hormone JA undergoes a rhythm oscillator as well. The clock regulates JA levels with a peak in the midday [69]. In *Arabidopsis*, the clock protein CCA1 are able to bind to the JA biosynthetic gene promoter *LIPOXYGENASE2* [86]. The circadian clock can regulate the expression of the JA receptor *CORONATINE INSENSITIVE1*, as well as the expression of *MYC2*, a positive transcription factor of JA signaling. TIME FOR COFFEE (TIC) acts as a key determinant in circadian clock, also acts as a negative transcription factor in JA-mediated defense responses. It was found to inhibit the transcription of *CORONATINE INSENSITIVE1* to repress MYC2 protein accumulation [87].

## 3. Plant Immune Responses Involving Salicylic Acid (SA) and Jasmonic Acid (JA)

The plant hormone SA is produced in response to biotrophic pathogens invasion and induces SAR and JA is essential for the defense against necrotrophic pathogens and herbivorous insects [31]. Oscillation of hormones level plays a key role in proper coordination of immune response and plant development. The SA-signaling network involves transcription co-activators NPR1, NPR3, and NPR4. Whether NPR1 is a receptor of SA remains unclear [88,89].

### 3.1. Role of SA and JA in Coordinating Plant Immune Response and Growth

Recent chromatin immunoprecipitation studies indicate that morning-phased *CCA1*, evening-phased *TOC1*, and *PPR* family genes can combine with hundreds of gene promoters [59,86,90,91,92]. Circadian clock-related elements are regulated directly by plant hormones such as SA and jasmonic acid (JA), not by chance [93]. Maximal JA accumulation occurs around midday, whereas SA peaks around midnight [69]. *ISOCHORISMATE SYNTHASE 1* (*ICS1*) is a direct clock target gene that encodes a key enzyme involved in SA biosynthesis. *ICS1* expression is controlled by *CCA1 HIKING EXPEDITION* (*CHE*), a transcription factor of evening-phased elements [65]. *ICS1* gene expression, like SA accumulation, peaks at midnight and is involved in pathogen infection in the morning.

Morning-phased *LHY* and *CCA1* positively regulate resistance against oomycete pathogen *Hpa* and bacterial pathogen *P. syringae* [62,87], whereas evening-phased *TOC1* negatively regulates resistance against bacteria. With SA treatment in the subjective morning or evening under 3 h constant light, defense-related genes showed increased expression mainly 3 h after morning SA application [42], whereas genes related to growth and development showed increased expression mainly after evening SA application. SA treatment has a consistent effect on redox rhythm, and morning application has a positive effect on plant defense [87,94], whereas evening application has a negative effect. Such a shift of plant defense during the course of a day may reflect adaptation to changing diurnal physical conditions; temperature and humidity in the morning are typically more conducive to pathogen challenge [64]. The clock system also regulates plant defenses via other clock-related genes such as phosphate transporter gene *PHOSPHATE TRANSPORTER 4;1* (*PHT4;1)*, which *is* regulated by *CCA1*, a negative regulator of SA signaling [95]. SA functions in both local defense against biotrophs and as an essential signal in distal organization for broad-spectrum SAR [33].

A functional JA signaling pathway is required for oscillation in susceptibility to the *fungus Botrytis cinerea* [68]. In *Arabidopsis*, SA produced during pathogen infection strongly antagonizes both JA biosynthesis and JA-responsive gene expression, resulting in a large set of JA-responsive genes, including *LIPOXYGENASE2* (*LOX2*), *VEGETATIVE STORAGE PROTEIN (VSP)*, and *PLANT DEFENSIN 1.2* (*PDF1.2)* down regulated [93]. Plants may show negative responses to inappropriate stimuli; e.g., inappropriate induction of immunity at night may inhibit plant growth. Experimental SA treatment of plants under constant dark conditions helped maintain stronger circadian rhythm, but caused a major reduction of fresh weight. SA treatment also had an inhibitory effect on the expression of various defense-related aquaporin genes. An aqueous environment is generally essential for pathogen virulence. Thus, timing of SA-mediated immune responses in the morning helps avoid conflict between SA immune response and growth-related activities that require water transport at night [96].

### 3.2. Is NPR1 an SA Receptor?

Recently, some data clearly pointed out that NPR1 has high-affinity to bind with SA [97]. Increasing evidence strongly supports the idea that NPR1 plays a key role in SA signaling and as an SA receptor.

#### 3.2.1. Molecular Structure of NPR1

The *NPR1* gene was first discovered through genetic screening of *Arabidopsis* mutants showing defects of SA-mediated pathogen resistance [24,25,91,98,99]. NPR1 was cloned as a novel protein containing a bipartite nuclear localization signal, two protein–protein interaction domains, N-terminal bric-a-brac, tramtrack, and BTB/POZ (broad-complex, tramtrack, and bric-à-brac/poxvirus, zinc finger) domains, and an ankyrin-repeat domain with diverse functions [100,101,102]. The BTB/POZ domain is found in proteins that function as substrate adapters of CUL3-based ubiquitin ligases for the degradation of specific substrates [103].

#### 3.2.2. NPR1 Signal Transduction Pathway

Both increase of endogenous SA levels and exogenous SA application lead to gene transcription that is greatly altered under pathogen induction [104]. NPR1, a transcription co-activator known as a “master” regulator of plant immunity, triggers change of SA level. NPR1 function requires direct combination of the transition metal copper with SA, similar to the requirement of phytohormone ethylene for copper [105]. The SA/NPR1 complex is sensitive to chelation by Ethylene Diamine Tetraacetic Acid (EDTA), and the combination between SA and NPR1 could not be detected until testing of equilibrium dissociation constant (Kd) for incubation of SA with NPR1, coupled to the solid phase. The C-terminus of NPR1 contains two Cys residues (Cys521, Cys529) that play essential roles in binding SA to NPR1 [88].

The above findings clearly indicate that NPR1 is the SA receptor, and that they bind directly to each other. Partially reduced NPR1 oligomers become monomers under pathogen infection, and this process triggers SA accumulation (Figure 1) [33]. In the absence of SA or pathogen challenge, NPR1 is degraded by the proteasome and its inhibitory effect on effector-triggered cell death and anti-pathogen defense is eliminated [84]. For this, NPR1 requires Cullin-3 (CUL3)-based E3 ubiquitin ligase, a ligand with significant binding activity that can combine BTB domain-containing proteins [100,106,107,108]. NPR1 does not interact directly with CUL3, and it has, therefore, been proposed that association of CUL3 with NPR1 may be mediated by other adaptors that have the same domain as NPR1 [109]. NPR3 and NPR4, homologues of NPR1 that contain BTB domains, are good candidate adaptors [84,110]. *npr3-1*/*npr4-3* double mutant showed enhanced level of *PR* gene expression regardless of pathogen induction, and reduced susceptibility to both *P. syringae* pv. *maculicola* (*P.s.m*.) ES4326 and *Hyaloperonospora parasitica* Noco2. NPR1 protein stability thus makes the *npr3-1*/*npr4-3* double mutant insensitive to SAR induction. Binding affinity to SA is higher for NPR4 (*K*_d_ = 23.54 ± 2.742 nM) than for NPR3 (*K*_d_ = 176.7 ± 28.31 nM) [84,88,111]. Pathogen induction results in generation of an SA gradient whereby NPR1 accumulates in adjacent cells and promotes cell survival and SA-mediated resistance [112]. NPR1 may be degraded by NPR3 or NPR4 under stress, with suppression of programmed cell death during ETI at an infection site [23,89]. Surprisingly, the latest study shows that NPR1 functions separately with NPR3/NPR4. NPR3/NPR4 and may bind to SA at low levels to repress defense gene expression, the transcription co-activator NPR1 will promote defense gene expression on the accumulation of SA. Although both NPR1 and NPR3/4 are SA receptors, they function differently in transcriptional regulation of SA-induced defense gene expression, a NPR1 mutant loss the function in binding SA, it promotes SA-induced defense gene expression, whereas a NPR4 mutant represses SA-induced immune responses [97].

These findings, taken together, demonstrate the essential role of NPR1 as a receptor of SA, and also negative NPR3 and NPR4 controlling protein levels of NPR1 in plant immunity [84].

### 3.3. Plant Defense Network Is Controlled by Cytosolic NPR1

NPR1 as a key regulatory factor in the cross-talk between SA and JA signaling. Interaction between NPR1 and transcription factor TGA1 roles in the SA signaling, nuclear localization of NPR1 is not required for a plant defense network between SA and JA signaling, and therefore, the network is modulated by the NPR1 stays in cytosol [113]. Once the necrotrophic pathogens *Alternaria solani* and *Botrytis cinerea* are infected, the SA pathway will be manipulated through NPR1 and two JA-dependent defense genes expression including *PROTEINASE INHIBITORS I* and *II* will be suppressed for disease symptom development [68].

In brief, both JA and SA are of vital importance in plant immunity and thus coordinates the plant defense as well as growth regulation through various mechanisms.

## 4. Central Role of NPR1 in SA-Mediated Plant Defense Response

NPR1 is a key regulator of the SA-signaling network. SA induces degradation of NPR1, and NPR1 activates *PR1* transcription by combining with TGA factors following the transfer to the nucleus [33], as detailed in this section.

### 4.1. Role of NPR1 in Plant Immunity

Plant evolution involves continuous improvement and sophistication of immune systems. As has been talked above (2.2), *NPR1* shows defects in SA-mediated pathogen resistance on *Arabidopsis* mutants when it was first discovered; studies during the past two decades have clarified the role of NPR1 as a key transcriptional regulator affecting a wide variety of host genes involved in immune responses and biocompatibility [27,109,114]. This concept is supported by NPR1 overexpression experiments in *A. thaliana*. The conserved function of NPR1 has also been demonstrated in crop species such as rice, wheat, tomato, and apple [115]. Overexpression experiments with *Arabidopsis NPR1* (*AtNPR1*) homologues [116] in these species have revealed powerful resistance against fungal and bacterial pathogen attacks [117,118,119].

Two-hybrid screening studies of a yeast model have shown that NPR1 interacts with TGA transcription factors through its ankyrin repeat domain [120], and also acts as a transcription co-activator to regulate SAR gene expression. SA-mediated activation of *PR* genes is required for binding activity of NPR1 and TGA transcription factors to promoter elements [121,122]. TGA/OBF family members containing bZIP can also interact with NPR1 [120,121,123,124]. Following the binding of essential transcription factors to as-1 component, the *PR* gene promoter begins transcription and expression to initiate hypersensitive responses. NPR1 clearly functions as a key positive regulator in SA signaling, as well as a negative regulator in JA signaling [113].

### 4.2. Transfer of NPR1 Degradation Products to Nucleus

Under anti-pathogen defense response or SA induction, NPR1 is converted into monomers, which can be combined with TGA2 transcription factor to promote *PR* gene expression, and are released (transferred) from cytoplasmic sites to the nucleus (Figure 1) [24,25,70,99,125].

NPR1 negatively regulates programmed cell death. Occurrence of ETI in pathogen-infected cells leads to high SA accumulation, followed by NPR1 regulation of its target genes, CUL/ NPR3-mediated NPR1 turnover, and activation of cell death. CUL3/ NPR4-mediated degradation of NPR1 around infected cells is necessary to avoid unnecessary activation of resistance [70]. This event must occur prior to NPR1 regulation of its target genes, because NPR1 is a negative regulator of cell death by CUL3/ NPR4. Cys156 is the essential amino acid residue for NPR1 oligomerization. NPR1 can be assembled into a high-molecular weight oligomer in cytoplasm through S-nitrosylation among its Cys156 residues [126,127]. S-nitrosoglutathione (GSNO) facilitates Cys156 disulfide bonding and promotes NPR1 oligomer formation (Figure 1). Following SAR induction, oligomers in cytoplasm release large amounts of NPR1 monomers which are transferred to the nucleus and subsequently turned over [128].

NPR1 monomer is preferentially degraded relative to NPR1 oligomer, and in *35S::NPR1-GFP* plants, NPR1-GFP oligomer’s being converted to monomer is accelerated [70,102,128]. In the absence of new protein synthesis, the NPR1-GFP level is reduced or unchanged. In the presence of dexamethasone (DEX), it promotes nuclear translocation of a NPR1 fused to a DEX-responsive glucocorticoid receptor (NPR1-GR) and rapidly degraded [102,128]. In the absence of DEX, NPR1-GR stays localized mainly in cytoplasm. Constitutive degradation of the NPR1 monomer by proteasomes occurs only in the nucleus [128].

### 4.3. Differences between Systemic Acquired Resistance (SAR)-Induced and SAR-Uninduced Cells

NPR1 is a key regulator of the SA-signaling network during SAR development. Although SAR has important functions in plant immunity, preventing untimely SAR activation is essential for normal growth and development. CUL3-mediated degradation of NPR1 monomer prevents SAR activation in plants not under pathogen challenge [128]. In SA treatment experiments with *35S::NPR1-GFP* plants, NPR1 Ser11/15 phosphorylation level was greatly increased in SA-treated plants, whereas nuclear levels of phosphorylated endogenous NPR1 and NPR1-GFP were low in SA-nontreated plants. Turnover of phosphorylated NPR1 is essential for activation of SAR, and unphosphorylated NPR1 may be degraded in combination with target gene promoters. Gene transcription may be restricted by proteasomes, as a result of the destruction of transcription factors or proteolysis of co-activators, to protect assembly and installation of active transcription complexes [128]. Once the proteasome mediates NPR1 degradation, that SAR will respond stronger but more transiently [43].

### 4.4. Role of NPR1 in Circadian Clock Core Loop

NPR1 monomer level has recently been shown to peak at night, suggesting that rhythmic translocation of NPR1 to nucleus, controlled by rhythmic oscillation of endogenous SA, may regulate circadian clock genes. NPR1 antagonistics both SA and *P. syringae* infection in the regulation of circadian rhythms. The amplitude reduction and phase delay triggered by a transient SA treatment is prevented by NPR1, the amplitude reduction induced by *Pst* DC3000 infection was also enhanced in *npr1-1* mutant plants [83]. NPR1 may function as an intrinsic regulator of clock gene *TOC1* in response to rhythmic accumulation of endogenous SA [69]. NPR1 is most likely not the only regulator of *TOC1*. Treatment with 1 mM SA on 3-week-old soil-grown wild-type or on *NPR1-3* mutant plants does not significantly alter the rhythm cycle. TOC1 expression level is increased immediately by SA induction [42]. Reduction of TOC1 level shortens the rhythm cycle, whereas increase of the TOC1 level lengthens the cycle [71,129]. Other clock genes such as *LHY*, in contrast to *TOC1*, are sensitive to SA induction. The regulatory mechanism of clock genes in relation to plant immunity was elucidated by experiments using *LHYp:LUC* reporter (*LHYp (TBSm):LU*C). *LHY* expression peaked at dawn, while *TOC1* expression was delayed by SA treatment, because *LHY* is an antagonist of *TOC1* in the clock. NPR1 will strengthen the circadian clock when the redox rhythm is disrupted by challenges, through regulation of both morning-phased *LHY* and evening-phased *TOC1* [69].

## 5. Plant Immunity and Redox Rhythm as Related to the Circadian Clock

Redox rhythm is closely associated with plant immunity, and both are related to the circadian clock. Early studies of the fungus *Neurospora crassa* suggested regulation of cellular redox state by the circadian clock [130]. Daily timekeeping functions are driven by both transcription-translation feedback loops (circadian clock) and non-transcriptional redox oscillations in numerous organisms, including *Arabidopsis* [2,3,4]. Nicotinamide adenine dinucleotide oxidation state (NAD/NADH ratio) also presents a daily rhythm. 

### 5.1. Reactive Oxygen Species (ROS) Oscillation Triggers Plant Immunity

The alteration of cellular redox state is one of the earliest occurring responses in a challenged cell [131]. ROS (e.g., H_2_O_2_) and reactive nitrogen species (RNS; e.g., nitric oxide [NO]) have closely related roles in plant immunity [132,133,134]. Plant immune responses are effectively triggered only if ROS and RNS signaling are activated simultaneously. The balance between ROS and RNS production is crucial in determining the fate of infected cells, and the success of plant immune responses [133,135].

Morning-phased *CCA1* controls ROS homeostasis by regulating expression of catalases (e.g., catalase mutant *CAT2*) involved in plant defense, Alvina et. al., pointed that *CCA1* is a master regulator of ROS homeostasis (Figure 2) [35]. In *Arabidopsis*, inhibition of *CAT2* activity by SA leads to increased H_2_O_2_ level (Figure 2) and subsequent sulfenylation (oxidation of thiol -SH into sulfenic acid -SOH) of tryptophan synthetase β-subunit 1 (TSB1) [136]. In contrast to historical and multigenerational circadian clocks based on differing transcription-translation feedback loops, the peroxiredoxin active site (responsible for redox rhythms) is conserved in a wide variety of organisms, indicating a possible origin of redox-based circadian clocks [2,96].

ROS, which are generated during photosynthesis, respiration, and other metabolic processes, and peak around midday, are toxic and can interfere with plant growth and development (Figure 2). In association with plant physiological responses, the level of redox species (which scavenge ROS) peaks at around the same time. Oscillation of peroxide-scavenging enzyme (antioxidant) levels is upregulated close to subjective dawn (Figure 2). ROS production can be inhibited by peroxide-scavenging enzymes (Figure 2). Circadian clock may, therefore, regulate ROS-related genes to prevent changes in ROS homeostasis [137,138]. Experiments on an *Arabidopsis* model to test the hypothesis that the antioxidant network is regulated by the circadian clock at the transcriptional level revealed that the ROS homeostasis regulon transcription was altered in *CCA1* and *LHY* mutants. Evening-phased elements in promoters of antioxidant genes involved in transcriptional regulation were found to mitigate oxidative stress (Figure 2). These findings support the concept that central oscillators control oxidative stress through transcriptional regulation [35,139].

### 5.2. NADP/NADPH Oscillation Reflects Redox Rhythm

Circadian redox rhythm in anucleate red blood cells (erythrocytes) is reflected in oscillations of both oxidation state of ROS scavenger peroxiredoxins and NADP/NADPH levels, both of which lack transcription/translation activities [3,96]. Redox rhythm is reflected mainly in regular changes of redox coenzymes NADPH and NADP^+^. Zhou et al. [42] found that, under constant light, NADPH peaked before subjective dawn while NADP^+^ peaked before subjective dusk. Redox rhythm was sensitive to external perturbation; e.g., SA treatment disrupted NADPH/NADP^+^ rhythms and altered their proportions to induce expression of defense-related genes.

### 5.3. Oscillation of NPR1 Regulation Pathway

Because the redox rhythm is regulated by NPR1, circadian redox rhythm in plants is coupled with a genetic clock. This coupling is mediated by cellular redox changes induced by endogenous SA level, which induce degradation and rhythmic nuclear translocation of NPR1 [69,70]. Endogenous SA levels oscillate in a circadian pattern [69]. Reduction by SA and glutathione of ethyl ester (GSHmee), a redox-altering reagent, enhanced expression of evening-phased *TOC1* in an NPR1-dependent manner [140]. NPR1 is a general redox sensor that modulates clock gene *TOC1*. NPR1 monomer level peaks at night and displays a circadian oscillation pattern. Basal expression of *TOC1* was reduced in both an *NPR1* mutant and in an SA mutant [100]. Supplemental SA increased *TOC1* expression in the SA mutant but not in the *NPR1* mutant, regardless of treatment duration [42].

Redox signaling in plants helps determine the most suitable defense against pathogen infection, and the circadian clock helps predict various potential/ imminent attackers [132,141,142]. Upon pathogen challenge in *Arabidopsis*, stomatal closure is induced by NADPH oxidase RESPIRATORY BURST OXIDASE HOMOLOGUE D (RBOHD) [143,144]. SA accumulation is induced by RBOHF, another NADPH oxidase [145]. Expression of SA-dependent defense genes is induced by ROS produced by apoplastic peroxidases PRX33 and PRX34 [146,147,148,149]. Both stomatal closure and defense gene induction required NO, produced by NO synthase-like activity [96,150,151,152]. NO functions as a redox regulator to promote rhythmic nuclear translocation of NPR1, and thus plays an important role in the modulation of plant immunity [153].

Reduced glutathione (GSH) and oxidized glutathione (GSSG) form an essential antioxidant buffer system. Reduction of the GSH/GSSG ratio during pathogen infection leads to increased oxidative stress. NPR1 reduction does not lead to an immediate increase of the GSH/GSSG ratio [70]. GSH is, therefore, not a direct reducing agent of NPR1, although it has some correlation with redox changes of NPR1 and GSH/GSSG ratio. Other redox-related molecules such as glutathione and thioredoxin may be involved in NPR1 reduction. Pathogen challenge creates an SA gradient which is associated with changes in cellular redox and reduces the activity of two cytosolic thioredoxins (TRX-3h, TRX-5h) that catalyze monomerization of NPR1 and prevent its repolymerization (Figure 1) [126].

## 6. Future Prospects

We have summarized here a variety of evidence for connections between circadian clock, plant immunity, and redox rhythm. Cellular redox state is regulated by circadian clock. SA treatment of plants disrupted oscillations of NADPH/ NADP^+^ and altered their proportions, thereby triggering expression of defense-related genes. ROS have a toxic (biocidal) effect on invading bacteria, but can also injure host cells. NPR1 may be the core element that links together the circadian clock and redox rhythm. Plant defense systems are associated with the circadian clock for regulation of daily physical activity via a complex network system. Defense responses create an environment conducive to plant survival, but often lead to growth inhibition, thus setting up a perpetual competition for resource allocation. Our understanding of plant immunity and the circadian clock has undergone major, sequential progress during the past decade, and research in both areas is well advanced. On the other hand, SA is the last of the major plant hormones whose receptors have an intricate control of SA responses. Future studies will further elucidate the mechanism of how SA receptors regulate plant defense responses and the core elements that coordinate the circadian clock with plant immunity.

## Figures and Tables

**Figure 1 ijms-20-01211-f001:**
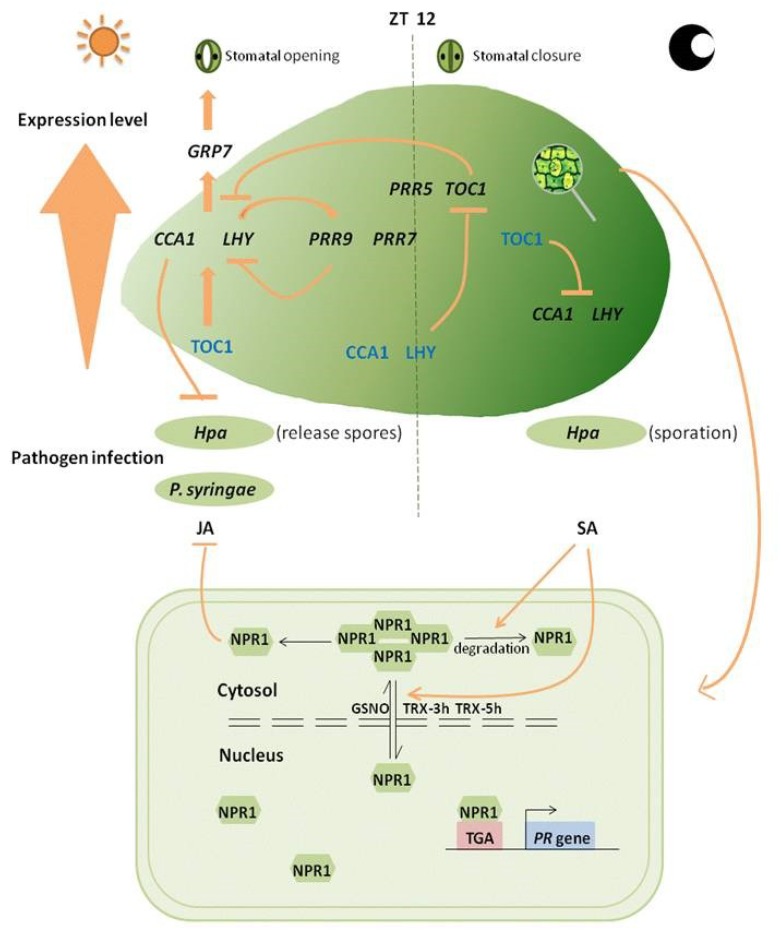
Circadian clock genes changes affect plant defense.

**Figure 2 ijms-20-01211-f002:**
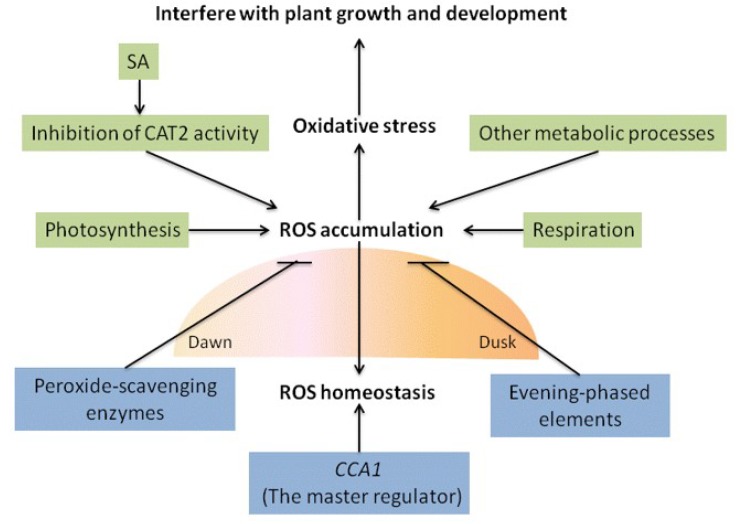
Reactive oxygen species (ROS) homeostasis. ROS, which are generated during photosynthesis, respiration, and other metabolic processes, peak around midday and are toxic to interfere with plant growth and development. *CCA1* is a master regulator of ROS homeostasis. In *Arabidopsis*, inhibition of *CAT2* activity by SA leads to increased H_2_O_2_ level. Oscillation of peroxide-scavenging enzyme levels is upregulated close to subjective dawn and inhibits ROS production. Evening-phased elements in promoters of antioxidant genes involved in transcriptional regulation were found to mitigate oxidative stress.

**Table 1 ijms-20-01211-t001:** Core loop genes are direct regulators of plant defense.

Gene Name	*CCA1*		*LHY*		*TOC1*
**Expression phase**	Morning		Morning		Evening
**Clock phenotypes**	Shorter periods		Shorter periods		Shorter periods
**Relationship to core loop**	Negatively regulates *TOC1*, *LHY* and *CCA1* gene expression		Negatively regulates *TOC1*, *LHY* and *CCA1* gene expression		Repressor directly binding to *CCA1* and *LHY* promoters in early night
**Expression influencing factors**	Affected by flg22, RPP4, and infection by *P. syringae*, *H. arabidopsidis*, and *B. cinerea*		Increased by SA		Increased by SA and suppressed by mutations disrupting *ICS1*, *NPR1*, or ROS signaling
**Types**	Mutants	Overexpression	Mutants	Overexpression	Mutants
**Susceptibility**	More susceptible to *P. syringae*, *H. arabidopsidis*, and *B. cinerea*	More susceptible to *P. syringae*, *T. ni* More resistant to *H. arabidopsidis*	More susceptible to *P. syringae*, *H. arabidopsidis*, and *B. cinerea*	More susceptible to *P. syringae*	Higher SA-induced resistance to *P. syringae*
**References**	[35,47,51,57,61,62,64,65,66,67,68]		[35,47,51,57,61,62,65,68]		[13,35,42,45,52,58,59,65,69,70,71,72]

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
