# Peer review of "NPR1 and Redox Rhythm: Connections, between Circadian Clock and Plant Immunity"

_ijms, 2019, doi:10.3390/ijms20051211_

Reviewer 1 Report

Authors well written NPR1 and Redox rhythm. No further claim.

Author Response

According to the comments of other reviewers, we have revised the full text in details.

Reviewer 2 Report

The review by Zhang et al tries to summarize recent evidence for the interaction between NPR1, Redox rhythms and the circadian clock. I found it difficult to connect the different headings in a streamline manner, as it was not well prepared. The review is a compilation of results which sometimes gets monotonous to read. It lacks integration into concepts of physiology or mechanism.

It’s hard to find out from the figure when the clock activates and inhibits itself and the SA and JA pathway and why and how ?. 

TOC1 is mentioned to repress CCA1/LHY in text but I could not see it in figure. Also, at line 103 page 4 the text does not reflect in figure. i would expect the figure to be more precise and self explanatory.

The figure legend is missing so it is difficult to track what authors are saying

The organization of the review, which is of course a matter of personal preferences. It would make more sense if authors discuss first what is pathogenesis and how NPR1 and redox state regulate the output of pathogenesis, followed by a brief introduction of circadian clock and finally how these pathways integrates. I feel it’d make it easier to read and understand.

A model depicting role of SA in pathogenesis will be good to see. Also, if authors are not explaining JA in detail it can be omitted or put a paragraph about it.

Other minor points

2)      Line 44: EFR and FLS2 not defined

3)      Lines 109: elf3 is not defined

4)      Line 123: PCD not defined

5)      Line 303: NPR1-3 is gene or mutant?

Author Response

Point 1: It’s hard to find out from the figure when the clock activates and inhibits itself and the SA and JA pathway and why and how ?.

Response 1: Details have been added.

Point 2: TOC1 is mentioned to repress CCA1/LHY in text but I could not see it in figure. Also, at line 103 page 4 the text does not reflect in figure. i would expect the figure to be more precise and self explanatory.

Response 2: Details have been added.

Point 3: The figure legend is missing so it is difficult to track what authors are saying

Response 3: The legend of figure 1 has been attached. We also added figure 2 to make it easier to track.

Point 4: The organization of the review, which is of course a matter of personal preferences. It would make more sense if authors discuss first what is pathogenesis and how NPR1 and redox state regulate the output of pathogenesis, followed by a brief introduction of circadian clock and finally how these pathways integrates. I feel it’d make it easier to read and understand.

Response 4: Thanks for your advice.

Point 5: A model depicting role of SA in pathogenesis will be good to see. Also, if authors are not explaining JA in detail it can be omitted or put a paragraph about it.

Response 5: The details about JA have been attached (Line 192-197, Line 200, Line 206, Line 229-232, Line 289-298).

Point 6: Line 44: EFR and FLS2 not defined

Response 6: Line 46- The definitions have been added.

Point 7: Lines 109: elf3 is not defined

Response 7: Line 117- The definitions have been added..

Point 8: Line 123: PCD not defined

Response 8: Line 132- The definitions have been added.

Point 9: Line 303: NPR1-3 is gene or mutant?

Response 9: Line 326- “mutant” has been added behind “NPR1-3”

Reviewer 3 Report

This manuscript provides a comprehensive revision of the molecular mechanisms involved in coordination of plant clock and defense responses. Authors reviewed the role of core clock components on the regulation of plant immunity, the molecular mechanisms underlying NPR1 regulation of plant defenses, and the roles of SA and ROS on the regulation of clock and defense mechanisms.

While the overall topics covered in this revision are appropriate, in most passages the information is confusing and difficult to follow. This is in part because the flow of information is often incoherent but also because of improper grammar, English usage and, sometimes, sentence meaning. The manuscript needs very serious editing to solve these issues (some specific ones are pointed below). In addition, some inaccurate statements/sections (pointed below) should be revised.

Major points:

Line 31- “The circadian clock…physiological reaction” (what do authors mean by: the clock “detects” the precise time)

Line 32- “ Furthermore..dark” (the clock does not “have” and endogenous rhythm but rather “drives” endogenous ryhtms)

Line 37-References 10-12 are not the right ones.

Line 39- “In addition,…oscillation” (what do authors mean by: “will function in the normal state through preserves the balance”?. Also here it is inaccurate to state that 6% of the genes are clock controlled as more recent and more comprehensive transcriptomics analyzes have shown that there are many more)

Line 47- “..pathogens, possible??..”

Line 50 – “appeared??”

Line 57 – “.. can reflect ROS generation” (what do authors mean by reflect?)

Line 59 – what is/are the references for the sentence finishing in this line?

Line 67- “ Coordination…essential??..to maximize development..” what reference supports this statement

Line 80 – What are clock conditioning models?

Line 87-88 – Mutations in LHY also do not disrupt the clock. Why this was not mentioned?

Line 104-105- There is a discrepancy for the CCA1OX phenotypes mentioned in the text and table 1

Lines 111-112- what is the reference for “Circadian clock regulates R genes as well as PAMP perception genes”

Line 130- What do authors mean by “circular lighting?”

Lines 127-140 (Section 2.2.2) indicate that CCA1 has a role in regulating non-clock related activities. The arguments in this section, that some phenotypes are evident in CCA1-OX plants but not LHY, do not support this idea. It is possible that CCA1 and LHY have different target genes and that their role in regulating the core clock function may be different than their role in regulating clock outputs. If this were the case, both functions would be “clock related”.

Lines 160-161 – What is the reference that supports that the link between clock and defense “plays an important role in metabolism and development”?.

Line 174- what do authors mean by “evening-phased clock”?

Line 221- the sentence “No notable SA-binding activity by NPR1 occurs in Arabidopsis.” Is confusing. Do authors mean that SA does not bind to NPR1 in the absence of pathogen challenge?

Lines 236-238. This sentence is confusing.

Line 264- NPR1 is not “degraded” into monomer (may use “converted”)

Lines 281-283- Incorrect statement. Dexamethasone does not promote the fusion of NPR1 to the GR, but rather promote nuclear translocation of a NPR1-GR protein fusion.

Lines 295-297. Sentence is confusing

Lines 298-311 (section 4.4). This paragraph completely neglected another study (ref 75) that investigated the role of NPR1 in clock regulation.

Lines 318-319. Confusing statement.

Line 329- What do authors mean by “multigenerational”?

Author Response

Response to Reviewer 3 Comments

1: Line 31- “The circadian clock…physiological reaction” (what do authors mean by: the clock “detects” the precise time).

Response 1: Line 33-Inserted “to help plants make right reaction at the right time” after “physiological reaction”.

Point 2: Line 32- “Furthermore...dark” (the clock does not “have” and endogenous rhythm but rather “drives” endogenous ryhtms).

Response 2: Line 34-“has” was replaced by drives.

Point 3: Line 37-References 10-12 are not the right ones.

Response 3: Line 38-References 10-12 were replaced by others that support our point (Line 423-431).

Point 4: Line 39- “In addition,…oscillation” (what do authors mean by: “will function in the normal state through preserves the balance”?. Also here it is inaccurate to state that 6% of the genes are clock controlled as more recent and more comprehensive transcriptomics analyzes have shown that there are many more).

Response 4: For the sake of understanding, we replaced “In addition,…oscillation” with “In addition, at least 30% genes expression with with robust oscillations in Arabidopsis are associated with physiological rhythms driven by circadian clock.” according to reference 15(also replaced by recent analyzed).

Point 5: Line 47- “..pathogens, possible??..”

Response 5:Line 49- “triggered by...cell death in the plant” replaced by “which is triggered by specific recognition between pathogen effectors and immune receptors in host cell, generally encompasses a programmed cell death at the site of pathogen recognition known as hypersensitive response (HR).” Line 52- deleted “effectors can also be recognized by cytoplasmic”

Point 6: Line 50 – “appeared??”

Response 6: Line 55- Deleted “appeared, is triggered by the hypersensitive responses”, insert “occurred following HR”

Point 7: Line 57 – “.. can reflect ROS generation” (what do authors mean by reflect?)

Response 7: Line 63- Deleted “can reflect ROS generation”, inserted “has been evidenced as the source of ROS”. Also reference 39 changed.

Point 8: Line 59 – what is/are the references for the sentence finishing in this line?

Response 8: Line 66-67- References has been added.

Point 9: Line 67- “ Coordination…essential??..to maximize development..” what reference supports this statement.

Response 9: Line 74- The sentence was replaced by “Although the coordination between biotrophic pathogens and host may related to circadian clock, but it had never been firmly established about the link between plant immunity and the circadian clock, also the coordination provided a schedule for plant innate immunity to deal with exogenous pathogens” . Also reference 45 changed.

Point 10: Line 80 – What are clock conditioning models?

Response 10: Line 89- The complete sentence was deleted.

Point 11: Line 87-88 – Mutations in LHY also do not disrupt the clock. Why this was not mentioned?

Response 11: Line 96- Inserted “LHY(lhy)”

Point 12: Line 104-105- There is a discrepancy for the CCA1-OX phenotypes mentioned in the text and table 1

Response 12: Table 1- “More resistant to” replaced “and”

Point 13: Lines 111-112- what is the reference for “Circadian clock regulates R genes as well as PAMP perception genes”.

Response 13: We’re sorry for that the reference didn’t be found, for the sake of accuracy, we deleted the sentence.

Point 14: Line 130- What do authors mean by “circular lighting?”

Response 14: Line 160- Inserted “(12 hr light/ 12 hr dark)”behind “circular lighting”.

Point 15: Lines 127-140 (Section 2.2.2) indicate that CCA1 has a role in regulating non-clock related activities. The arguments in this section, that some phenotypes are evident in CCA1-OX plants but not LHY, do not support this idea. It is possible that CCA1 and LHY have different target genes and that their role in regulating the core clock function may be different than their role in regulating clock outputs. If this were the case, both functions would be “clock related”.

Response 15: Line 158- The first sentence was replaced by “Whether CCA1 plays a role in regulation of non-clock-related activities for enhancement of anti-pathogen defense responses?”   Line 162- reference 45 was added behind the second sentence. Line 167- “These findings suggest...independently” was replaced by “These findings suggest that the circadian clock regulate plant innate immunity as an output under both light conditions”.

Point 16: Lines 160-161 – What is the reference that supports that the link between clock and defense “plays an important role in metabolism and development”?

Response 16: Line 198-199- The sentence was deleted.

Point 17: Line 174- what do authors mean by “evening-phased clock”?

Response 17: Line 213- Turned “clock” to “elements”.

Point 18: Line 221- the sentence “No notable SA-binding activity by NPR1 occurs in Arabidopsis.” Is confusing. Do authors mean that SA does not bind to NPR1 in the absence of pathogen challenge?

Response 18: Line 266- The sentence was deleted. Line 264- added “(Figure 1)” behind “accumulation”.

Point 19: Lines 236-238. This sentence is confusing.

Response 19: Line 282- The original sentence was replaced by “Although both NPR1 and NPR3/4 are SA receptors, they function differently in transcriptional regulation of SA-induced defense gene expression, a NPR1 mutant loss the function in binding SA, it promotes SA-induced defense gene expression, whereas a NPR4 mutant represses SA-induced immune responses”

Point 20: Line 264- NPR1 is not “degraded” into monomer (may use “converted”)

Response 20: Line 323- Turned “degraded” to “converted”.

Point 21: Lines 281-283- Incorrect statement. Dexamethasone does not promote the fusion of NPR1 to the GR, but rather promote nuclear translocation of a NPR1-GR protein fusion.

Response 21:  Line 340- The sentence “NPR1 undergoes fusion to DEX-responsive glucocorticoid receptor (NPR1-GR) and is degraded rapidly in nucleus [96].” was replaced by “it promotes nuclear translocation of a NPR1 fused to a DEX-responsive glucocorticoid receptor (NPR1-GR) and rapidly degraded [96, 123].”.

Point 22: Lines 295-297. Sentence is confusing

Response 22: Line 356- The sentence was replaced by “Once the proteasome mediates NPR1 degradation that SAR will response stronger but more transient [42].”.

Point 23: Lines 298-311 (section 4.4). This paragraph completely neglected another study (ref 75) that investigated the role of NPR1 in clock regulation.

Response 23: Line 363- Insert “NPR1 antagonistics both SA and P. syringae infection in the regulation of circadian rhythms. The amplitude reduction and phase delay triggered by a transient SA treatment is prevented by NPR1, the amplitude reduction induced by Pst DC3000 infection was also enhanced in npr1-1 mutant plants [75]”.

Point 24: Lines 318-319. Confusing statement.

Response 24: Line 384- The sentence was deleted.

Point 25: Line 329- What do authors mean by “multigenerational”?

Response 25: Line 405- Added “historical and” in front of “multigenerational”.

Round  2

Reviewer 2 Report

The review by Zhang et al : NPR1 and Redox Rhythmx: Connections, Between Circadian Clock and Plant Immunity has been improved from previous version. Just a simple suggestion to the authors is that please don't take any task casually i can still find some mistakes in the manuscript (typos and abbreviations) which are not major but its annoying to read from the reviewers point of view, as they are also putting their efforts to make it better. There are several typos which i am not mentioning here so please check it carefully. Also, try to reframe the sentences carefully so as it is understandable to readers and not just to avoid plagiarism.

here are some points as follows

1. Line 39 "with with" should be "with"

2. Line 45-46 abbreviations are not correct "FLAGELLIN‐SENSING 2 (EFR) and EF‐Tu receptor (FLS2)" should be the other way around i.e. FLAGELLIN‐SENSING 2 (FLS2) and EF‐Tu receptor (EFR)

3. Line 180 Peakin ? should be "Peak in"

4. Line 181 circadian clock can regulate of the JA receptor CORONATINE INSENSITIVE1 expression should be circadian clock can regulate the expression of JA receptor CORONATINE INSENSITIVE1

5. Line 183 sentence needs to be reframed JA‐mediated defense responses daily oscillation required for the clock associated protein TIC interacts with MYC2

6. Line 216 Please abbreviate VSP and PDF1-2

7. Description of Figure 2 is missing in text

8. Line 377 Alvina et. should be Alvina et. al.,

Author Response

Point 1: Line 39 "with with" should be "with"

Response 1: Line 39, it has been corrected.

Point 2: Line 45-46 abbreviations are not correct "FLAGELLIN‐SENSING 2 (EFR) and EF‐Tu receptor (FLS2)" should be the other way around i.e. FLAGELLIN‐SENSING 2 (FLS2) and EF‐Tu receptor (EFR)

Response 2: Line 45-46, we’ve reversed the position of “FLAGELLIN‐SENSING 2” and “EF‐Tu receptor”.

Point 3: Line 180 Peakin ? should be "Peak in"

Response 3: Line 180, it has been corrected.

Point 4: Line 181 circadian clock can regulate of the JA receptor CORONATINE INSENSITIVE1 expression should be circadian clock can regulate the expression of JA receptor CORONATINE INSENSITIVE1

Response 4: Line 190, it has been revised.

Point 5: Line 183 sentence needs to be reframed JA‐mediated defense responses daily oscillation required for the clock associated protein TIC interacts with MYC2

Response 5:Line 191, the sentence was changed to “TIME FOR COFFEE (TIC) acts as a key determinant in circadian clock, also acts as a negative transcription factor in JA-mediated defense responses. It was found to inhibit the transcription of CORONATINE INSENSITIVE1 to repress MYC2 protein accumulation”.

Point 6: Line 216 Please abbreviate VSP and PDF1-2

Response 6: Line 221, PHT4;1 Line 228, VSP and PDF1.2. The full name and abbreviation of these 3 genes have been completed.

Point 7: Description of Figure 2 is missing in text

Response 7: It has been added

Point 8: Line 377 Alvina et. should be Alvina et. al.,

Response 8: Line 392, it has been corrected.

Reviewer 3 Report

The manuscript has improved from its previous version.

Still, the article is difficult to read in several passages and there are plenty grammatical mistakes that I think should be corrected.

Here I list a few findings:

Line 45, should be “ (FLS2)” not ( EFR)

Line 46, should be “(EFR)” not ( FLS2)

Lines 68-71, “Although the coordination between biotrophic pathogens and host may related to circadian clock , but it has never been firmly established about the link between plant immunity and the circadian clock , also the coordination provides a schedule for plant innate immunity to deal with exogenous pathogens”. This sentence is really confusing, should be revised.

Line 87, What lhy mutant do authors refer to?

Line 98, should be CCA1 not CC   A1

Line 123, “CCA1 encodes transcription factors” should be singular not plural

Lines 129-130, “indicating the ability of circadian clock to control large numbers of PAMP perception genes even in the absence of pathogen attacks”. The first part of this sentence does not support this statement.

Author Response

Point 1: Line 45, should be “ (FLS2)” not ( EFR)

Response 1: Line 45, it has been corrected.

Point 2: Line 46, should be “(EFR)” not ( FLS2)

Response 2: Line 46, it has been corrected.

Point 3: Lines 68-71, “Although the coordination between biotrophic pathogens and host may related to circadian clock , but it has never been firmly established about the link between plant immunity and the circadian clock , also the coordination provides a schedule for plant innate immunity to deal with exogenous pathogens”. This sentence is really confusing, should be revised.

Response 3: Line 69-75, the sentence was replaced by “The coordination between biotrophic pathogens and host provides a daily schedule for plants to defense exogenous pathogens, that plant defense may related to circadian clock. However, it has never been firmly established about the link between them”.

Point 4: Line 87, What lhy mutant do authors refer to?

Response 4: Line 95, the reference has been added.

Point 5: Line 98, should be CCA1 not CC   A1

Response 5: It has been corrected.

Point 6: Line 123, “CCA1 encodes transcription factors” should be singular not plural

Response 6: Line 131, it has been revised to singular.

Point 7: Lines 129-130, “indicating the ability of circadian clock to control large numbers of PAMP perception genes even in the absence of pathogen attacks”. The first part of this sentence does not support this statement.

Response 7: Line 137, the sentence has been changed to “indicating the ability of circadian clock to regulate the host immune response”.